# Assessment Heartworm Disease in the Canary Islands (Spain): Risk of Transmission in a Hyperendemic Area by Ecological Niche Modeling and Its Future Projection

**DOI:** 10.3390/ani13203251

**Published:** 2023-10-18

**Authors:** Rodrigo Morchón, Iván Rodríguez-Escolar, Ricardo E. Hernández Lambraño, José Ángel Sánchez Agudo, José Alberto Montoya-Alonso, Irene Serafín-Pérez, Carolina Fernández-Serafín, Elena Carretón

**Affiliations:** 1Zoonotic Diseases and One Health GIR, Biomedical Research Institute of Salamanca—Research Center for Tropical Diseases University of Salamanca (IBSAL-CIETUS), Faculty of Pharmacy, Campus Miguel Unamuno, University of Salamanca, 37007 Salamanca, Spain; 2Internal Medicine, Faculty of Veterinary Medicine, Research Institute of Biomedical and Health Sciences, University of Las Palmas de Gran Canaria, 35001 Las Palmas de Gran Canaria, Spain; alberto.montoya@ulpgc.es (J.A.M.-A.); elena.carreton@ulpgc.es (E.C.); 3Biodiversity, Human Diversity and Conservation Biology Research Group, Campus Miguel Unamuno, University of Salamanca, 37007 Salamanca, Spain; ricardohl123@usal.es (R.E.H.L.); jasagudo@usal.es (J.Á.S.A.); 4Instituto Universitario de Enfermedades Tropicales y Salud Pública de Canarias, Universidad de La Laguna, Avda. Astrofísico Fco. Sánchez, S/N, 38203 La Laguna, Spain; ireneser@ull.edu.es (I.S.-P.); cfserafin@ull.edu.es (C.F.-S.)

**Keywords:** *Dirofilaria immitis* risk model, ecological niche model, forward projection, *Dirofilaria immitis*, *Culex* spp., Canary Islands, Spain

## Abstract

**Simple Summary:**

Cardiopulmonary dirofilariasis is a vector-borne zoonotic disease caused by the nematode parasite *Dirofilaria immitis*. This disease is cosmopolitan and mainly affects canids and felids and accidentally humans. The Canary Islands (Spain) are considered an endemic area and some of its islands such as the islands of Tenerife and Gran Canaria (provincial capitals) are considered hyperendemic. The aim of our study was to develop a quantitative proposal of the risk of infection by *D. immitis* using an ecological niche model (ENM), taking into account environmental and bioclimatic variables that affect the presence of culicid mosquitoes, to maximize its resolution and, in addition, estimate the impact of future climatic conditions in 2080 on the risk of infection. Nineteen environmental and bioclimatic variables were processed and modeled using ArcMap 10 and MaxEnt 3.4. The model was weighted by the number of *Dirofilaria* spp. generations and validated with the prevalence and geolocation of *D. immitis*-infected dogs across the Canary Islands. The risk map of *D. immitis* infection in the Canary Islands is high in all the islands, except for the higher altitude areas, and as for the analysis of range change, the projection for 2080 does not show a large increase in the extension of areas suitable for *Culex* spp., but rather a modification in the distribution of these areas. This model will help veterinary and public health professionals to carry out more effective and localized prevention and control of cardiopulmonary dirofilariosis (heartworm disease).

**Abstract:**

Heartworm disease is a vector-borne zoonotic disease caused by *Dirofilaria immitis*. The Canary Islands (Spain), geolocated close to the coast of Western Sahara, is an archipelago considered hyperendemic where the average prevalence in domestic dogs is high, heterogeneous, and non-uniform. In addition, *Culex theileri* has been reported as a vector of the disease on two of the most populated islands. Our aim was to develop a more accurate transmission risk model for dirofilariosis for the Canary Islands. For this purpose, we used different variables related to parasite transmission; the potential distribution of suitable habitats for *Culex* spp. was calculated using the ecological niche model (ENM) and the potential number of generations of *D. immitis*. The resulting model was validated with the geolocation of *D. immitis*-infected dogs from all islands. In addition, the impact of possible future climatic conditions was estimated. There is a risk of transmission on all islands, being high in coastal areas, moderate in midland areas, and minimal in higher altitude areas. Most of the dogs infected with *D. immitis* were geolocated in areas with a high risk of transmission. In 2080, the percentage of territory that will have been gained by *Culex* spp. is small (5.02%), although it will occur toward the midlands from coastal areas. This new model provides a high predictive power for the study of cardiopulmonary dirofilariosis in the Canary Islands, as a hyperendemic area of the disease, and can be used as a tool for its prevention and control.

## 1. Introduction

Cardiopulmonary dirofilariosis, caused by *Dirofilaria immitis* Leidy, 1856 (Chromadorea, Onchocercidae), is a zoonotic disease that mainly affects dogs and cats, where humans can be infected, being considered as an accidental host [1,2]. It is a vector-borne disease where its vectors are culicid mosquitoes of different genus (*Culex* spp., *Aedes* spp., *Anopheles* spp., among others). Its distribution is cosmopolitan, located mainly in tropical and subtropical regions, although it is expanding into areas with cooler average temperatures that were previously free from the disease [2,3,4].

The presence of vectors and water, to which their life cycle is linked, climatic and environmental factors, and the location of microfilariae reservoirs are factors to be taken into account when talking about the existence of the disease in a given area. In addition, the period in which parasite molting occurs in the vector (from microfilariae to Larva 3) is shortened when the outside temperature increases and vice versa, slowing down at temperatures below 14 °C [4,5,6].

Climate change is causing alterations in the risk of vector-borne disease transmission in animal and human populations [7]. Rising temperatures as well as seasonal changes in precipitation and the impact of the human geographic footprint, among others, directly contribute to the survival and active presence of culicine mosquitoes over longer periods in a given area [3,7,8]. In Europe, the disease, traditionally endemic in southern countries (Portugal, Spain, France, Italy, Greece and Turkey), has spread toward the center-north and is currently in continuous expansion [3]. In Spain, cardiopulmonary dirofilariosis is an endemic disease with a mean prevalence in dogs of 6.47% [9], a seroprevalence in cats of 9.4% [10], and where there are reports of infected wild animals and humans [3,11]. The Canary Islands (Spain), which are geolocated close to the coast of Western Sahara, is the Spanish autonomous community with the highest average prevalence (11.58% in dogs). However, the prevalence per island is very heterogeneous, being 15.65% in La Palma, 0% in El Hierro, 11.54% in La Gomera, 17.32% in Tenerife, 16.03% in Gran Canaria, 1.74% in Fuerteventura, and 0.99% in Lanzarote [9]. In addition, one of the species transmitting the disease, *Cx. theileri* Theobald, 1903, has been detected in Gran Canaria and Tenerife [12] as well as *Cx. pipiens* Linnaeus, 1758, a vector transmitting the disease in the Iberian Peninsula [13,14].

In terms of disease control, one of the most important tools is the creation of infection risk maps [3]. One of the methodologies applied for this is the use of geographic information systems (GISs) based on temperature records with which, in some cases, the number of *Dirofilaria* spp. generations (the complete development of *Dirofilaria* spp. L3 in mosquito vectors during 30 days) [5,15,16,17,18,19,20,21,22,23] and, in others, the humidity, rainfall, irrigated areas [24,25], and the ecological niche model (ENM) incorporating all of these as variables and other ecoclimatic variables [26] to create risk models.

However, we are hardly aware of any studies in relation to dirofilariosis and their vectors. It is known that among the most important factors for the transmission of the disease are the optimal environmental conditions for the development of insect vectors. Insects are highly dependent on climate to activate their metabolism and behavior. For example, in hematophagous species, temperature, precipitation, or wind speed play an important role in reproduction, abundance, survival, and activity. In this regard, the inclusion of a variable describing the spatial distribution patterns of its main vector in risk models will greatly improve the model accuracy.

Taking into account the health relevance of the zoonotic diseases and the apparent current expansion of vector insects, largely favored by anthropogenic alterations (such as the aforementioned global warming), it is necessary to develop detailed analyses that focus on the environmental circumstances that promote them. In this sense, spatially explicit risk models based in correlations between the presence of zoonotic diseases and the variables associated with their transmission can be extrapolated to other territories where data do not exist and to other timeframes to be able to anticipate their dynamics of dispersion and take precautionary measures to mitigate their negative effects. Thus, the objective of this study was to develop a more accurate risk model of *D. immitis* transmission in the Canary Islands. For this purpose, we used a set of key variables associated with parasite transmission: the potential distribution of suitable habitats for *Culex* spp., and the potential number of *D. immitis* generations. The incorporation of suitable habitats for *Culex* spp. in the risk model represents a novel contribution to substantially improve the predictive capacity of existing studies.

## 2. Materials and Methods

### 2.1. Study Area Description

The Canary Islands (27°37′ and 29°25′ N, 13°20′ and 18°10′ O) are an archipelago located in the Atlantic Ocean 1400 km from the Iberian Peninsula and 97 km from Morocco and Western Sahara (Figure 1). It is the southernmost and westernmost Autonomous Community of Spain and one of the outermost regions of the European Union.

The archipelago covers an area of 7492 km^2^ and is made up politically of seven islands with their own administration: El Hierro, La Gomera, La Palma, and Tenerife make up the province of Santa Cruz de Tenerife, and Fuerteventura, Gran Canaria, and Lanzarote make up the province of Las Palmas. In relation to their topography, the Canary Islands are an archipelago of very recent volcanic origin, only 30 million years old. They present several phases of lava flows with the typical volcanic relief, giving rise to a very mountainous orography in which the Teide (Tenerife) stands out, the highest peak, not only of the islands, but of all of Spain with an altitude of 3715 m. On each of the Canary Islands, there are three different areas: the mountains; the midlands, located immediately below the peaks; the coasts. The western islands (Tenerife, La Palma, Gomera, and Hierro together with Gran Canaria) have the highest relief, while the eastern islands (Lanzarote and Fuerteventura) have the lowest altitude [27].

The Canary Islands have a subtropical oceanic climate, with mild temperatures all year round due to the influence of the sea, and in summer due to the trade winds. In terms of rainfall, there are significant variations. For example, on the island of La Palma, in some areas of the island, annual rainfall can exceed 1200 mm. On the eastern islands, rainfall is lower than in the west, with Fuerteventura and Lanzarote having an arid semi-desertic climate. Regarding the Köppen Climate Classification, the eastern islands, mainly Lanzarote and Fuerteventura, have a predominance of the BWh climate, hot desert, characterized by hot or very hot summers with mild winters and low rainfall. The BSh and BSk climates, warm semi-arid and cold semi-arid, respectively, are found on all of the islands, normally substituting the desert climates as the altitude increases. In Tenerife, they are found in the coastal area from northwest to southeast and in the lower midlands from southeast to northeast. The temperate climates, Csa and Csb, where precipitation exceeds evaporation, are found mainly in the temperate zone and even in some of the intertropical zones. The Csa corresponds to the typical Mediterranean climate, with a warm summer, and the Csb to the oceanic Mediterranean with a mild summer. In the highest areas of Tenerife, the Teide, the Dfc climate is found, which is cold without a dry season and a cool summer [28].

### 2.2. Data

#### 2.2.1. *Culex* spp. Collection

We used occurrence points of *Cx. pipiens* and *Cx. theileri* from all islands from the data obtained by the Canary Islands Entomological Surveillance of the Instituto de Enfermedades Tropicales y Salud Pública de Canarias, Melero-Alcíbar et al. [29,30], Morchón et al. [12,31], the Spanish Government [32], and Serafín-Pérez et al. [33], together with distributional records for the species from GBIF [34] for the Canary islands due to the few records of the species from this archipelago, these being the only *Culex* spp. species on the islands dated and transmitters of the disease in the Canary Islands and Spain to date [12,13,14]. Occurrence data were processed at a spatial resolution of 1 km^2^ to avoid biases associated with the spatial autocorrelation of aggregate occurrence records.

#### 2.2.2. Environmental and Bioclimatic Data

The bioclimatic data, 19 bioclimatic variables related to temperature and precipitation, were obtained from the WorldClim website [35] at a spatial resolution of an approximately 1 km^2^ spatial resolution corresponding to the current data (1970–2000) and predicted for the 2040s, 2060s, and 2080s [36]. Between the 19 bioclimatic variables, a multicollinearity test was performed in R software using Pearson’s correlation coefficient [37] to avoid autocorrelation between variables. Variables with a correlation coefficient of r > ±0.75 were discarded. The bioclimatic variables chosen, according to vector biology, were mean annual temperature (°C) (BIO_1_), isothermia (BIO_3_), seasonality of temperature (SD × 100) (BIO_4_), mean temperature of the wettest quarter (°C) (BIO_8_), mean temperature of the driest quarter (°C) (BIO_9_), annual precipitation (mm) (BIO_12_), and seasonality of precipitation (coefficient of variation) (BIO_15_). In addition, due to their influence on vector distribution, variables such as human footprint [38], presence of irrigated crop areas, location of artificial and natural bodies of water [39], shrub density, and herbaceous density [40] were used. The selected variables are shown visually in Appendix A.

*Culex* spp. data as well as bioclimatic and environmental variables were processed at a 1 × 1 km resolution, at the same extent and with the GCS_WGS_1984_coordinates using ArcMap 10.8.

### 2.3. Culex spp. Ecological Niche Model

The distribution of habitat suitability for *Culex* spp. was calculated using ecological niche models (ENMs). ENMs are ecoinformatic tools that set suitability values in the environmental space of species distribution based on the occurrence locations and environmental data [41,42,43,44,45]. Recent studies have applied the ENM methodology for different parasitic diseases and their vectors [46,47,48,49,50].

We used MaxEnt [51] to model the habitat suitability and potential geographical distribution of *Culex* spp. within the study area as the life cycle of both species was developed taking into account the same variables. To choose an appropriate amount of model complexity [52], we used the KUENM package [53] in R version 3.6.1 [54], which selects the best MaxEnt models of a series of candidates arranged by different combinations of parameter settings. In total, we created 119 candidate models for each species by combining the full set of independent variables, 17 values of regularization multipliers (0.1–1.0 at intervals of 0.1, 2–6 at intervals of 1 and 8 and 10), and the seven possible combinations of three entity classes (linear, quadratic, and product). The best model for each species was selected taking into account the statistical significance (partial ROC < 0.05), omission rates (E = 5%), and model complexity (models within two units of the minimum value among candidate models considering the Akaike information criterion corrected for small sample sizes (AICc). For each species, we then created the final best-fit model using all occurrence records and the selected parameterization for each species (Appendix A) and projected it to the study area.

### 2.4. Dirofilaria Immitis Generations

The number of annual generations of *D. immitis* was calculated using the model described by Genchi et al. [15], Simón et al. [24], and Rodríguez-Escolar et al. [26] using the R program. This model states that the complete development of *D. immitis* in mosquito vectors (extrinsic incubation) requires the accumulation of 130 growing degree days (GDDs) within 30 days, the maximum life expectancy of the vector. A number of GDDs is accumulated per day equivalent to the number of degrees that the average daily temperature exceeds 14 °C. The threshold of 130 GDDs is only accepted if it is reached within 30 consecutive days. For the calculation of the number of generations of *D. immitis* on the different islands of the Canary archipelago, the mean daily temperature data [55] from 1990 to 2016 were used [56].

### 2.5. Dirofilaria Immitis Risk Map

A weighting between the final NEM for the *Culex* spp. and *D. immitis* generations was carried out, and a *D. immitis* risk map was generated using the spatial analysis tool of ArcMap 10.8 with the raster calculator [26].

To validate the *D. immitis* risk map, georeferenced points of *D. immitis*-infected dogs in the different islands of the Canary Islands were obtained [9] and superimposed on the risk map to see which areas they inhabited.

### 2.6. Forward Projection and Rank Change Analysis

To assess the possible impacts of future climate conditions (temperature and precipitation), we projected the best models calibrated in MaxEnt for each species to three time periods (2040s (2021 to 2040), 2060s (2041 to 2060), and 2080s (2061 to 2080) and three RCPs 8.5 scenarios using the HadGEM3-GC21-LL model [36].

Once the projections were generated, to establish changes in suitable habitats for *Culex* spp. in the Canary Islands, we transformed the ENM and future time projections into binary presence/absence maps using the 10 percentile training presence logistic threshold of the current model as a threshold. A range change analysis using the biomod2 package of the R software was applied to determine the territories where there will be a change in the distribution of *Culex* spp. on the islands as a consequence of climate change [57]. The analysis consisted of calculating the percentage of cells that gained or lost habitat suitability for the models projected to 2040, 2060, and 2080 compared to the current model.

## 3. Results

### 3.1. Culex spp. Habitat Suitability

An ecological niche model was generated for *Culex* spp. in the geographical area of the Canary Islands, which presented a curve value (AUC) of 0.810, indicating a good predictive power. Figure 2 shows the habitat suitability for *Culex* spp. with a maximum value of 0.77 (high suitability) and a minimum value of 0.041 (low suitability). The contribution of each of the variables in the *Culex* spp. ecological niche model is presented in Table 1 and shown visually in Appendix A.

The variables with the highest percentage contribution to the model were human footprint and shrub density, being 83.32% and 6.78%, respectively. For the rest of the variables, the values were less than 4.02%. According to the habitat suitability distribution map, the most suitable areas for *Culex* spp. are mainly located in coastal areas with a greater human presence, especially the islands of Tenerife, Gran Canaria, and Lanzarote and some urban areas of Fuerteventura and La Palma. Less populated areas, higher above sea level and with less presence of water, presented a lower habitat suitability, being the islands with less capacity to host *Culex* spp. La Gomera and El Hierro.

### 3.2. Number Dirofilaria Immitis Generations

The calculation of the number of generations of *D. immitis* in the Canary archipelago is represented as a map in Figure 3. The highest number of generations (>4.8) was found in the coastal areas, decreasing with altitude in each of the islands, with the exception of Lanzarote and Fuerteventura, where the number of generations was high practically over the whole territory due to the low altitude of these islands. Inland areas showed values of between two and three generations, with the highest areas such as Teide showing low values of one generation.

### 3.3. Potential Risk of Transmission of Dirofilaria Immitis

The infection risk map for *D. immitis* in the Canary Islands is shown in Figure 4. In general, the risk of transmission is high on all islands, with the exception of the higher altitude areas. Three ranges of values have been established (high, medium and low), according to which 30% of the territory of the archipelago is in an area with a high risk of transmission; 31% would be in the second range with a medium risk, being low in 38.9% of the territory. The areas with high risk values correspond to the coastal areas of the islands where there is a high human footprint, humidity, and irrigated crops. A medium risk of transmission is found in midland areas with a higher altitude and less human presence. Mountain areas, with more irregular orography and lower humidity, are locations with a low risk of infection.

To try to validate our transmission risk model, *D. immitis* infected dogs were superimposed on the model, which showed that 77% of these infected dogs were located in high transmission risk areas, 18% in medium risk areas, and 5% in low risk areas (Figure 5).

### 3.4. Future Projection

Regarding the range change analysis, projection into the future for the years 2040, 2060, and 2080, according to climate change scenario RCP 8.5, does not show a large increase in the extent of suitable areas for *Culex* spp. but rather a modification in the distribution of these areas (Figure 6). The percentage of territory gain for *Culex* spp. in 2040 is 5.23%, 1.22% in 2060, and 5.02% in 2080. However, in these periods, there is also a loss in the extent of suitable areas of 1.05%, 0.15%, and 5.02%, respectively. Thus, in the first two scenarios, there is a small increase in extent, while by 2080, the same percentage of territory is lost as is gained. These small increases in areas suitable for the vector are toward midland areas of higher altitude than the coast.

## 4. Discussion

This paper offers a quantitative proposal of the risk of infection by *D. immitis* for the Canary Islands (Spain) based on the use of ecological niche modeling tools and taking into account a large number of predictor variables. Several predictive models of *Dirofilaria* spp. infection risk in Europe have been built using GIS based only on temperature records to estimate the number of generations of the parasite that can develop extrinsically in the vector and, in some of them, the duration of disease activity throughout the year has also been considered [5,15,16,17,18,19,20,21,22,23]. In all of these studies, an oceanic climate with sufficient humidity for vector development was assumed for all of Western Europe, without taking into account regions with a more arid or subtropical oceanic climate such as the Canary Islands archipelago. For the Canary Islands, there was a previous study with a simpler methodology, where the number of generations of *Dirofilaria* spp. and the environmental and edaphic humidity were used as environmental variables with values for the whole of Spain, the peninsular part, and the Balearic and Canary Islands [24]. In contrast, our model for the Canary Islands is of high resolution and includes a large number of variables highly related to the main vectors of disease dispersal in the archipelago, thus providing greater geographical precision when establishing the risk of dirofilariosis on more local scales, allowing for more accurate extrapolations of risk to other territories. The very mechanics of our procedure also facilitate geographical inference of the variables considered. Thus, our results indicate that there is a possibility of the risk of infection (values always greater than 0 in an interval of 0 to 1) throughout the Canary Island archipelago.

To demonstrate the predictive power of our *D. immitis* infection risk model, an overlay of geo-referenced records of dogs infected by *D. immitis* (no data for other *Dirofilaria* spp. species known to date in the Canary Islands archipelago) on the different islands was performed [9], where most of them were in high and medium risk areas, indicating that our model can be optimal in assessing the risk of the disease.

Bioclimatic conditions, mainly temperature and humidity, have a great influence on the extent and seasonality of dirofilariosis, mainly due to the environmental conditions necessary for vector development [3]. For the Canary Islands, a high risk of infection is predicted in coastal areas with a larger human footprint and humidity, which decreases as the altitude increases (mid-altitude and mountain areas) or the humidity decreases. The extrinsic development of *D. immitis* larvae in vectors can occur in all islands throughout most of the Canary Islands, with rainfall being the main limiting factor for the development of mosquito populations. However, this circumstance can be locally compensated for by the presence of both natural and artificial water bodies, irrigated crops, or urban areas that act as heat islands, offering an ideal habitat for mosquito breeding [25,58,59,60]. Previous epidemiological studies revealed a prevalence of *D. immitis* in the canine population of 67.02% on the island of Gran Canaria [61], and 23–41.8% in Tenerife [62,63,64] until the application of widespread preventive chemotherapy, where the prevalences are currently 16.03% in Gran Canaria and 17.32% in Tenerife [9].

The Canary Islands, despite having a dry subtropical climate, have a high relative humidity due to the offshore trade winds [65]. This, together with the typical system of water collection and storage in open reservoirs and the large number of expected generations of the parasite, result in ideal local habitats for breeding mosquito populations. Due to these characteristics, the conditions for the transmission of *D. immitis* are favorable in many parts of the Canary Islands [24]. These circumstances have led to the high prevalence of canine dirofilariosis in the archipelago in previous studies [9]. The human footprint is the variable with the highest statistical value in our model, indicating that the areas with the highest risk of infection are those with a high population density and a high concentration of canals and open water storage structures. We can say that, according to our model, the risk will be high whenever high humidity, irrigated areas, or a high human footprint coincide in the territory.

The result of our future projections under climate change scenarios indicates minimal modifications in the current distribution area of *Culex* spp. The percentage of territory gain for these species in 2040 is 5.23%, 1.22% for 2060, and 5.02% for 2080. For oceanic/volcanic islands, due to their restricted area, a distribution shift is expected to occur for arthropods only along the altitudinal gradient. In our study, in the projection to 2040, a small tendency to shift toward the center of islands with higher latitudes seems to be observed. However, by 2060, the changes are very small and by 2080, the same percentage of territory is lost as is gained. This can be explained by a previous study in the Azores, which stated that climate change can have a strong impact on island-dwelling species and that they may lose all of their suitable climate space [66]. Another explanation may be the strong association of the mosquito vector with urbanization, as urban areas provide the constant temperature and humidity necessary for its survival [67] and therefore, despite climate change, mosquito populations could be comfortably maintained in areas with a large human footprint on islands with only minor modifications in their dynamics. Finally, the importance of additional climatological elements, especially precipitation, should be emphasized. Climate change phenomena do not only involve changes in temperature, but also in precipitation, which is essential for establishing mosquito breeding sites and larval habitats. As a recent study in the Canary Islands indicates, future scenario projections expect an increase in aridity, with the least severe changes in Lanzarote and Fuerteventura (already quite arid at present). The areas where the greatest change in aridity is expected are the southeastern slopes of Tenerife and Gran Canaria, mainly due to decreasing precipitation [68]. The increase in aridity on the islands as a result of climate change could explain why the territory of habitat suitability for *Culex* spp. will not increase significantly in the future as rainfall decreases, with mosquito populations becoming established in areas where they do have sufficient humidity for their survival (urban areas, irrigated crops, water collection ponds).

## 5. Conclusions

Due to the continuous expansion of zoonotic diseases, largely favored by anthropogenic climate change, and their health relevance to public health, it is necessary to carry out detailed studies to determine the environmental circumstances that promote them. Thanks to the NEM methodology, we can establish correlations between the presence of these diseases and biotic variables, which can also be extrapolated to other territories where there are no data and to other time scenarios, in order to predict their evolution in the future and establish preventive measures to deal with them before they occur.

According to our risk model, in the Canary Islands, the risk of infection is high on almost all of the islands, with the exception of the higher altitude areas. Bioclimatic variables, human footprint, artificial water bodies, shrub density, and temperature are delimiting factors to be taken into account, together with *D. immitis* generations. Future projections under climate change scenarios have allowed us to visualize minimal modifications in the current range of *Culex* spp. due to the loss of their suitable climatic space to urbanization that provides the constant temperature and humidity necessary for their survival.

This model will help veterinary and public health professionals to carry out more efficient and localized prevention and control of dirofilariosis, taking into account the specific situation of each population. Further studies are needed to comprehensively assess the risk of *D. immitis* infection at the local level in order to take precautions to avoid the potential spread of the disease.

## Figures and Tables

**Figure 1 animals-13-03251-f001:**
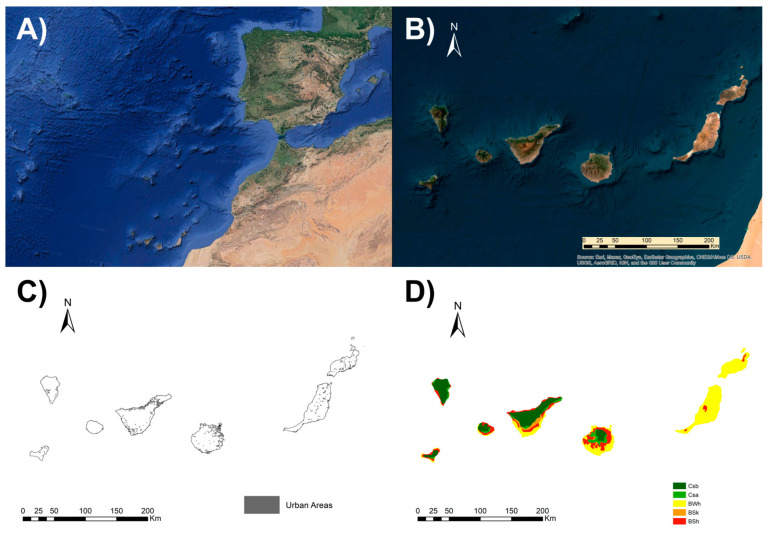
The Canary Islands (Spain) according to their: (**A**) localization, (**B**) orography, (**C**) location of human populations, and (**D**) climates according to the Köppen Climate Classification System (BSh: hot semi-arid climate; BSk: cold semi-arid climate; BWh: hot desert climate; Csa: hot-summer Mediterranean climate; Csb: warm-summer Mediterranean climate).

**Figure 2 animals-13-03251-f002:**
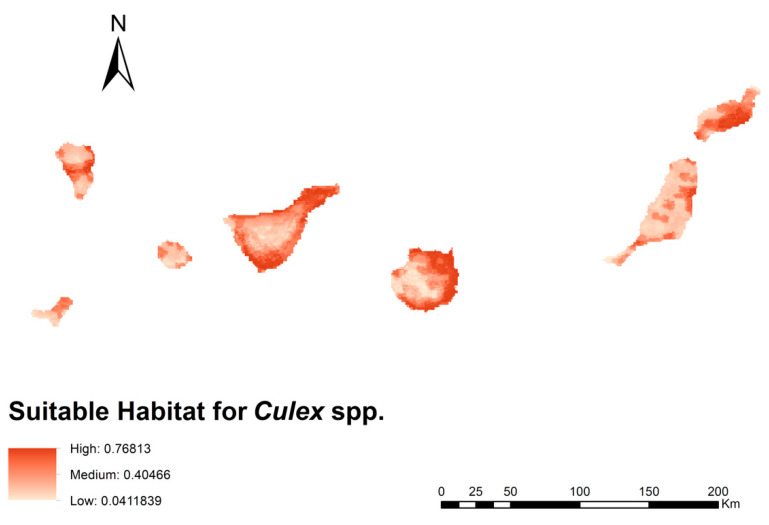
Ecological niche model for *Culex* spp. in the geographical area of the Canary Islands representing a suitable habitat. Parameters of the best model: multiplier regularization = 10 and feature classes = quadratic.

**Figure 3 animals-13-03251-f003:**
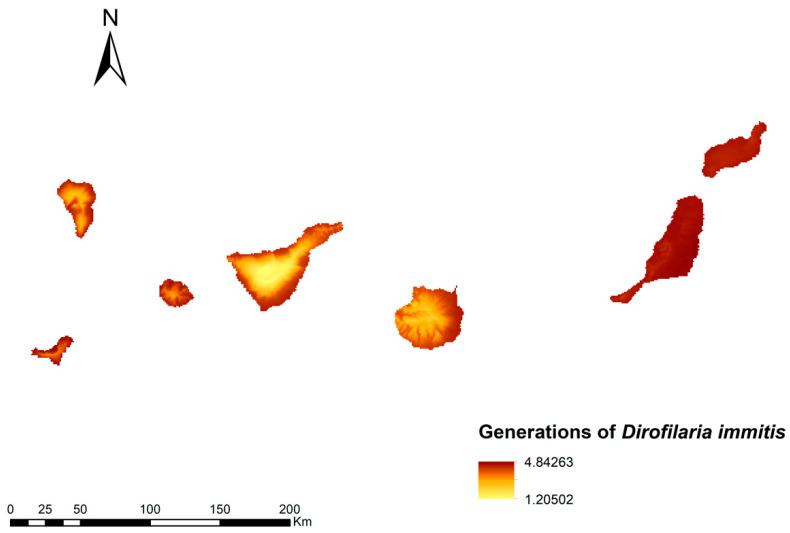
Prediction of the number of generations of *D. immitis* in the Canary Islands.

**Figure 4 animals-13-03251-f004:**
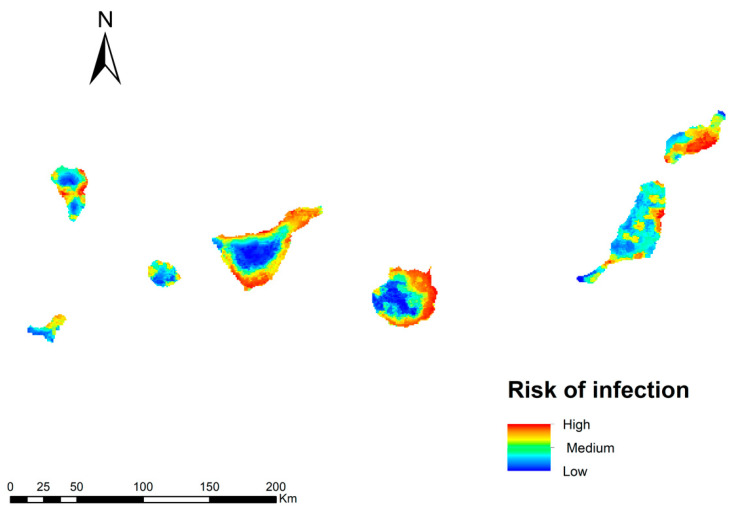
Model of the risk of *D. immitis* infection in the Canary Islands.

**Figure 5 animals-13-03251-f005:**
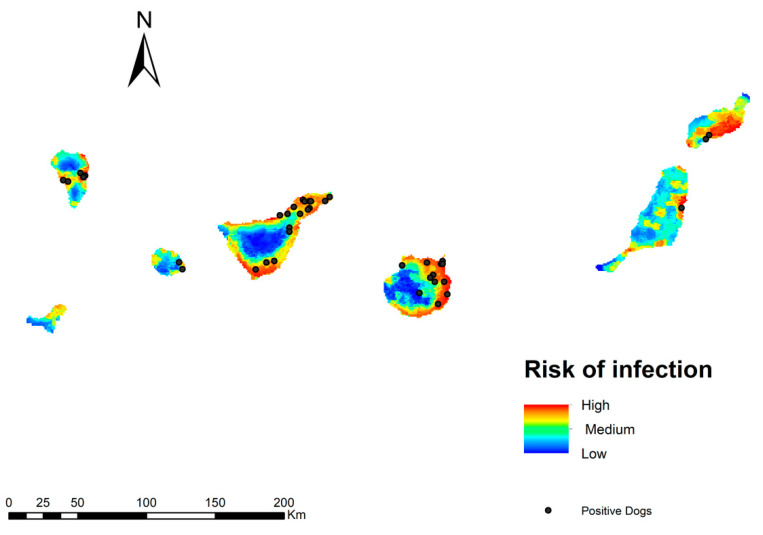
Model of the risk of *D. immitis* infection in the Canary Islands with the location of infected dogs according to Montoya-Alonso et al. [10].

**Figure 6 animals-13-03251-f006:**
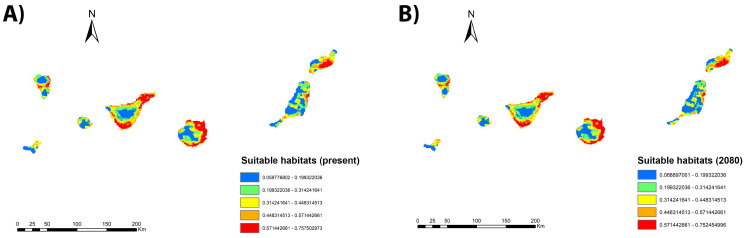
Suitable habitats for *Culex* spp. at present (**A**), and their projection into the future: 2080 (**B**) for the Canary Islands under climate change scenario RCP 8.5.

**Table 1 animals-13-03251-t001:** Analysis of the contribution of the 13 environmental and bioclimatic variables to the ecological niche model for *Culex* spp. in the Canary Islands.

Variable	Percent Contribution
Human footprint	83.32%
Shrub density	6.78%
Artificial water bodies	4.02%
BIO_4_ (temperature seasonality)	2.98%
BIO_8_ (mean temperature of wettest quarter)	1.36%
BIO_12_ (annual precipitation)	0.74%
Natural water bodies	0.61%
Irrigated crops	0.11%
BIO_3_ (isothermality)	0.05%
Herbaceous density	0.04%
BIO_9_ (mean temperature of driest quarter)	0%
BIO_15_ (precipitation seasonality)	0%
BIO_1_ (annual mean temperature)	0%

## Data Availability

Data used in the manuscript for tables and figures are presented in accompanying additional files.

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
