# Peer review of "Assessment Heartworm Disease in the Canary Islands (Spain): Risk of Transmission in a Hyperendemic Area by Ecological Niche Modeling and Its Future Projection"

_animals, 2023, doi:10.3390/ani13203251_

Round 1

Reviewer 1 Report

Cardiopulmonary dirofilariasis remains a public problem in many countries. This is a helminthiasis which affects canids and felids world wise. Moreover, it is also dangerous for humans, causing them pulmonary dirofilariasis. Therefore, research on distribution of this dangerous helminthiasis, and in particular its intermediate and final hosts, is always relevant. This study represents a fairly serious analysis of the parasitological situation of dirofilariasis in the Canary Islands based on ecological niche modeling. The manuscript undoubtedly makes a certain contribution to our knowledge about the functioning of the parasite-host system of Dirofilaria immitis.

The manuscript undoubtedly makes a certain contribution to our knowledge of the distribution of dirofilariasis, and undoubtedly meets the goals and objectives of the journal Animals and can be published.

But I have some remarks about this manuscript.

Great research, but the presentation of the results could have been better. Great research, but the presentation of the results could have been better. This is especially true for the design of the manuscript

Table and figures should be cited in the main text of the article after their first mention. I didn't find Table 1 and Figures 1-6 anywhere, which did not allow us to properly evaluate the manuscript.

Maybe it’s worth adding the word “assessment” to the article title?

Comment to the manuscript and section titles. In MDPI journals, in article and section titles, all words must begin with a capital letter – “Heartworm Disease in the Canary Islands (Spain): Risk of Transmission in a Hyperendemic Area by Ecological Niche Modeling and its Future Projection” or “Environmental and Bioclimatic Data”, etc.

I propose changing the title of subsection 2.1. like this –  “Study Area Description”;

and 3.2. Number of Dirofilaria immitis generations.

According International Code of Zoological Nomenclature (ICZN) at the first mention of genus or species (as in lines 53,80) its full Latin name with the author and year of description should be given; in relation all species (for example – Dirofilaria immitis (Leidy, 1856), Culex theileri Theobald, 1903, Culex pipiens Linnaeus, 1758, etc.). On subsequent mentions, the generic name is abbreviated to the first letter – C. theileri, C. pipiens

For the main “hero” of the article, the class and family names must also be given at the first mention – Dirofilaria immitis (Leidy, 1856) (Chromadorea, Onchocercidae)

And of course, remove unnecessary information here from the text after the list of references.

And some minor remarks:

Correct spelling of geographic coordinates (27°37′ and 29°25′ N, 13°20′ and 18°10′ W) (line 114).

Line 27 – “across the territory”?; may be better “across the Canary Islands” here?

Line 85 - usually write  larvae 3 – L3.

Line 133missing space.

Line 159 – please correct the error – Environmental.

Line 236 –        “… of each variables …”

Line 253 – Numbers in articles from 1 to 9 are written in words (one, two, three, etc.).

In Additional file 13 needs to be cleared of review mode.

The manuscript can be published, but corrections are needed.

English is fine, but you need to check the text for “bugs”

Author Response

Dear referee,

All suggested changes have been taken into account. We have responded point by point to all the referees' suggestions and they have been added to the text and marked in red. In addition, the manuscript has been reviewed by an English proofreading/editing service. The authors would like to thank him for your dedication and time spent in reviewing this manuscript.

We have also made improvements to the text to clarify it, and we have modified the simple summary, the abstract and the conclusions, taking into account the referees' suggestions.

Referee 1

Table and figures should be cited in the main text of the article after their first mention. I didn't find Table 1 and Figures 1-6 anywhere, which did not allow us to properly evaluate the manuscript.

Thank you for your comment. We have introduced the tables and figures into the manuscript. Animals edition had not done this. we have edited and introduced them to clarify the manuscript.

Maybe it’s worth adding the word “assessment” to the article title?

Comment to the manuscript and section titles. In MDPI journals, in article and section titles, all words must begin with a capital letter – “Heartworm Disease in the Canary Islands (Spain): Risk of Transmission in a Hyperendemic Area by Ecological Niche Modeling and its Future Projection” or “Environmental and Bioclimatic Data”, etc.

I propose changing the title of subsection 2.1. like this –  “Study Area Description”; and 3.2. Number of Dirofilaria immitis generations.

According International Code of Zoological Nomenclature (ICZN) at the first mention of genus or species (as in lines 53,80) its full Latin name with the author and year of description should be given; in relation all species (for example – Dirofilaria immitis (Leidy, 1856), Culex theileri Theobald, 1903, Culex pipiens Linnaeus, 1758, etc.). On subsequent mentions, the generic name is abbreviated to the first letter – C. theileri, C. pipiens

For the main “hero” of the article, the class and family names must also be given at the first mention – Dirofilaria immitis (Leidy, 1856) (Chromadorea, Onchocercidae)

And of course, remove unnecessary information here from the text after the list of references.

And some minor remarks:

Correct spelling of geographic coordinates (27°37′ and 29°25′ N, 13°20′ and 18°10′ W) (line 114).

Line 27 – “across the territory”?; may be better “across the Canary Islands” here?

Line 85 - usually write  larvae 3 – L3.

Line 133 – missing space.

Line 159 – please correct the error – Environmental.

Line 236 –        “… of each variables …”

Line 253 – Numbers in articles from 1 to 9 are written in words (one, two, three, etc.).

Thank you for your comment. We have made these changes to clarify the manuscript. We have included the persons who first identified the species and the year. However, the genus annotations for Culex are Cx. according to the International Code of Zoological Nomenclature.

In Additional file 13 needs to be cleared of review mode.

We have modified the table and sent it as a PDF to avoid the error.

The written language has been reviewed by native English scientists.

The written language has been reviewed by native English scientists.

Reviewer 2 Report

The work provides new data on the epidemiology of the parasitic dirofilariais disease; for this purpose, the authors created a risk model based on the distribution of Culex spp. vectors in correlation with various environmental and bioclimatic factors.

I am of an opinion that the article fits into scope of Animals and could be published after major corrections.

Comments:

1. When using a species name for the first time, please provide the common (if it exists) and scientific name with date and author(s), e.g. Dirofilaria immitis (Leidy, 1856).

2. Why was the model based only on the genus Culex, when the genera Anopheles and Aedes are also recorded in the Canary Islands? – taking into account also these vectors, the model would be more reliable.

3. Figures 1-6 and Table 1 are missing.

Minor editing of English language required.

Author Response

Dear referee,

All suggested changes have been taken into account. We have responded point by point to all the referees' suggestions and they have been added to the text and marked in red. In addition, the manuscript has been reviewed by an English proofreading/editing service. The authors would like to thank him for your dedication and time spent in reviewing this manuscript.

We have also made improvements to the text to clarify it, and we have modified the simple summary, the abstract and the conclusions, taking into account the referees' suggestions.

Referee 2

The work provides new data on the epidemiology of the parasitic dirofilariais disease; for this purpose, the authors created a risk model based on the distribution of Culex spp. vectors in correlation with various environmental and bioclimatic factors.

I am of an opinion that the article fits into scope of Animals and could be published after major corrections.

Comments:

  1. When using a species name for the first time, please provide the common (if it exists) and scientific name with date and author(s), e.g. Dirofilaria immitis (Leidy, 1856).

Thank you for your comment. We have made these changes to clarify the manuscript.

  1. Why was the model based only on the genus Culex, when the genera Anopheles and Aedes are also recorded in the Canary Islands? – taking into account also these vectors, the model would be more reliable.

The model was based on records of the presence of mosquitoes of the genus Culex as they are the only island species of this genus dated and transmitting the disease in the Canary Islands and Spain to date (Morchón et a., 2011, 2007, Bravo-Barriga et al., 2016).

Morchón, R.; Bargues, M.D.; Latorre-Estivalis, J.M.; Pou-Barreto, C.; Melero-Alcibar, R.; Moreno M. et al. Molecular Characterization of Culex theileri from Canary Islands.; Spain.; a potential vector of Dirofilaria immitisJ. Clin. Experiment. Pathol2011, S3, 001.

Morchón, R.; Bargues, M.D.; Latorre, J.M.; Melero-Alcíbar, R.; Pou-Barreto, C.; Mas-Coma, S.; Simón, F. Haplotype H1 of Culex pipiens implicated as natural vector of Dirofilaria immitis in an endemic area of Western Spain. Vector-Borne Zoonotic. Dis. 2007, 7, 653–658.

Bravo-Barriga, D.; Parreira, R.; Almeida, A.P.; Calado, M.; Blanco-Ciudad, J.; Serrano-Aguilera, FJ.; Pérez-Martín JE.; Sánchez-Peinado, J.; Pinto, J.; Reina, D.; Frontera, E. Culex pipiens as a potential vector for transmission of Dirofilaria immitis and other unclassified Filarioidea in Southwest Spain. Vet. Parasitol. 2016, 223, 173–180.

  1. Figures 1-6 and Table 1 are missing.

We have introduced the tables and figures into the manuscript. Animals edition had not done this. we have edited and introduced them to clarify the manuscript.

English is fine, but you need to check the text for “bugs”

The written language has been reviewed by native English scientists.

Round 2

Reviewer 2 Report

Dear Authors,

thank you for considering my previous comments; thank you also for the appropriate explanations.

I have one more comment - authors with dates in their scientific names are not always placed in the brackets. This type of issues is regulated by the International Commission on Zoological Nomenclature.

The correct spellings of scientific names are (without brackets !!):

Culex pipiens  Linnaeus, 1758

Culex theileri Theobald, 1903

Author Response

Thank you for your comments. We have corrected these errors and will take them into account for future manuscripts and their better understanding.
